# Gate-tunable Veselago interference in a bipolar graphene microcavity

Xi Zhang [1,10], Wei Ren[1,10], Elliot Bell [1], Ziyan Zhu [2,3], Kan-Ting Tsai [1], Yujie Luo [4,5], Kenji Watanabe [6], Takashi Taniguchi [7], Efthimios Kaxiras[2,8], Mitchell Luskin [9] & Ke Wang [1] ✉

The relativistic charge carriers in monolayer graphene can be manipulated in manners akin to conventional optics. Klein tunneling and Veselago lensing have been previously demonstrated in ballistic graphene pn-junction devices, but collimation and focusing efficiency remains relatively low, preventing realization of advanced quantum devices and controlled quantum interference. Here, we present a graphene microcavity defined by carefully-engineered local strain and electrostatic fields. Electrons are manipulated to form an interference path inside the cavity at zero magnetic field via consecutive Veselago refractions. The observation of unique Veselago interference peaks via transport measurement and their magnetic field dependence agrees with the theoretical expectation. We further utilize Veselago interference to demonstrate localization of uncollimated electrons and thus improvement in collimation efficiency. Our work sheds new light on relativistic single-particle physics and provide a new device concept toward next-generation quantum devices based on manipulation of ballistic electron trajectory.

The linear dispersion relationship of electrons in monolayer graphene[1,2] permits the manipulation of ballistic electron trajectories[3–6] in a manner akin to classical optics[7–14]. It has been demonstrated that an electron passing through a graphene pn-junction[15] can be collimated[16,17] through an angle-dependent Klein tunneling process[9,18–20] (analogous to a laser) and can also be refracted by a Veselago lensing process[21,22] (analogous to an optical lens) depending on the width and height of the pn-junction barrier. However, in state-of-the-art graphene pn-junctions, there still exist important technical challenges to create a full electronic version of advanced optical circuits. First, due to the relatively small pseudo-gap that can be created by an electrostatically-defined pn-junction, the Klein tunneling probability across the junction does not depend sensitively on the incident angle. In addition to electrons with perpendicular momentum to the junction that flow across it effortlessly, electrons with a finite incident angle $\theta$ relative to the perpendicular axis have a non-trivial transmission probability, limiting the resulting collimation efficiency. Moreover, the flatness of the charge neutrality boundary at which the charge carrier type switches in the pn-junction is highly sensitive to charge inhomogeneity (even in the highest-quality devices), introducing undesirable astigmatism analogous to a deformed lens. These limitations present significant challenges in studying more complex electron-optics processes such as controlled quantum interference, and in developing future quantum electronic devices based on more advanced electron-optical circuits.

[1]School of Physics and Astronomy, University of Minnesota, Minneapolis, MN 55455, USA. [2]Department of Physics, Harvard University, Cambridge, MA 02138, USA. [3]Stanford Institute for Materials and Energy Sciences, SLAC National Accelerator Laboratory, Menlo Park, CA 94025, USA. [4]Department of Electrical and Computer Engineering, University of Minnesota, Minneapolis, MN 55455, USA. [5]Department of Mechanical Engineering, University of Minnesota, Minneapolis, MN 55455, USA. [6]Research Center for Functional Materials, National Institute for Materials Science, Tsukuba, Ibaraki, Japan. [7]International Center for Materials Nanoarchitectonics, National Institute for Materials Science, Tsukuba, Ibaraki, Japan. [8]John A. Paulson School of Engineering and Applied Sciences, Harvard University, Cambridge, MA 02138, USA. [9]School of Mathematics, University of Minnesota, Minneapolis, MN 55455, USA. [10]These authors contributed equally: Xi Zhang, Wei Ren. ✉e-mail: kewang@umn.edu

In this work, we present a novel device architecture of a bipolar graphene microcavity that addresses these challenges with precise strain and electrostatic engineering. In this new device platform, we demonstrate novel electro-optics phenomena resulting from interference through consecutive Veselago lensing processes[23,24], referred to as "Veselago interference." We study the electrostatic and magnetic field dependence using low-temperature transport measurements and demonstrate quantitative agreement to relativistic single-particle physics. Finally, by utilizing the Veselago interference demonstrated here, we further localize uncollimated electrons to improve collimation efficiency, providing proof-of-concept demonstration of a new collimation scheme aimed toward future electro-optical devices.

## Results and discussion
### Sample preparation
A piece of monolayer graphene (MLG) is encapsulated by two layers of hexagonal boron nitride (hBN) using the standard dry-transfer technique[25-27]. The stack is subsequently transferred on top of two pre-patterned local bottom gates with a ~50 nm lateral gap, and an intentional 8 nm vertical height difference (Fig. 1a). Figure 1b shows an optical microscope image of the complete device. Atomic-force-microscopy data (Fig. 1c) across the 50 nm gapped region reveals that the device remains atomically flat on top of both gates as well as in the gap (see Methods), while lattice distortions are apparent at the boundaries of both gates (see Supplementary Note 2 and 17). We annealed the device at 350 °C for 5 min, so the strain is fully relaxed for the device region (cavity) in between the two barriers. The band structure of the graphene in and out of cavity remains pristine with linear dispersion relationship, expect for the barrier itself. This is an experimental design so that Klein tunneling and Veselago physics remain. The strain-induced band gap[28,29] effectively defines two tunnel barriers. A smooth strain over 8 nm is apparent in the AFM topography, resulting in an effective Klein barrier of net resistance 0.155 kΩ (see Supplementary Note 14), similar to that of previously-reported

electrostatically-defined Klein barriers[9]. While the Klein-tunneling angle dependence allows rough collimation of injected current, electrons with finite-incident angle can tunnel through the strain-defined barrier with low injection rate, making the regions in between two barriers an electronic analogy of an optical cavity.

### Observation of Veselago interference
For large (zero) incident angles, each strain-induced barrier provides near-perfect reflection (transmission) as expected from Klein tunneling angle dependence. For carriers with small but finite incident angle $\theta$, these two tunnel barriers effectively define a microcavity in between, where the injection rate to and loss rate from the cavity are equally low. The confinement of carriers with small incident angles is a meticulous experimental design to isolate, study, and manipulate those carriers of interest, allowing an interference loop to form via two consecutive Veselago refractions inside the cavity. In terms of application, these carriers are primarily responsible for the relatively low collimation efficiency previously reported. Further suppression of their contribution to electrical transport could be the key to improve collimation efficiency for future electron-optics devices.

The two local bottom gates, when biased at opposite voltages (of equal magnitude), electrostatically define a pn-junction symmetric against the charge neutrality boundary in the center of the cavity. The resulting carrier density distribution and the ballistic trajectory of carriers inside the cavity are shown in the color diagram of Fig. 1a, with a lower-bound estimation of electron mobility of ~90,000 cm$^2$ V$^{-1}$ s$^{-1}$ (see Supplementary Note 3). After being injected into the cavity with a small incident angle (and probability $p \ll 1$), each carrier inside the cavity first undergoes Veselago refraction as the carrier type changes in the middle of the device, gets reflected from the cavity boundary on the opposite side (with probability $1 - p \sim 1$), then undergoes a second Veselago refraction that ultimately brings the carrier back to its original position. At zero magnetic field, the charge interferes with itself. This results in an increase in measured resistance similar to weak

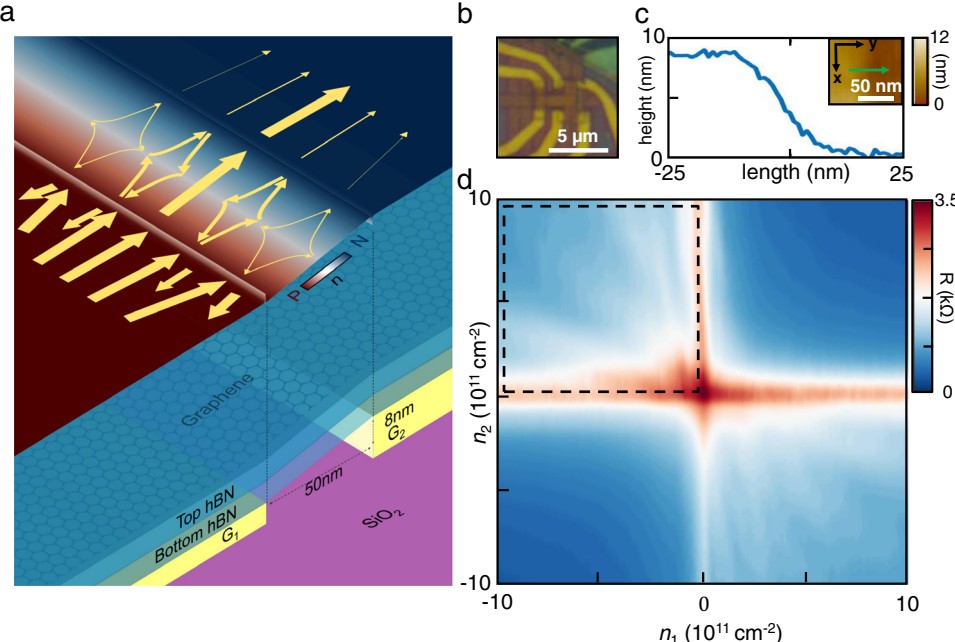

**Fig. 1 | Strain-defined and gate-tunable micro-cavity. a** (Bottom) hBN-encapsulated monolayer graphene transferred on top of a pair of atomically flat local bottom gates of different heights. (Top) The color plot maps the carrier density distribution in the device. Two narrow depletion regions form along the boundaries of the gates, where smooth lattice distortions happen. (Arrows) Ballistic trajectories of Veselago interference at $|n_1| = |n_2|$. **b** Optical image of a typical device. **c** AFM topography across the cavity. The 8 nm height difference and the smooth bending near the gate boundaries are visible. **d** The measured resistance as a function of the electron densities on Gate 1 ($n_1$) and Gate 2 ($n_2$), where resistance peaks are observed exclusively when $n_1$ and $n_2$ have opposite signs.

localization (Fig. 1d), but with two major differences. First, the effect is not weak, as all electrons trapped in the cavity form interference loops, independent of the specific incident angle. Thus, the cavity is also efficient in localizing all uncollimated electrons. Second, the effect can be turned on or off resonance by tuning inference paths with electrostatics, without the need for a magnetic field to suppress weak localization.

Figure 1d shows the measured resistance as a function of the densities of the two adjacent bottom gate regions, with $n_1$ and $n_2$ corresponding to the carrier density in the Gate 1 and Gate 2 regions, respectively. Resistance peak traces are observed only when the carrier densities in two regions are of opposite charge (when a pn-junction is formed within the cavity), ruling out the possibility for them to arise from disorder. Each data point on the interference peak, is contributed by a collection of Veselago interference loops with different individual incident angles and spatial phases, and lack overall coherence (see Supplementary Note 12). Absence of Fabry-Pérot interference (or lack of oscillatory dependence on carrier density) is expected and experimentally cross-checked (see Supplementary Note 13).

Figure 2a–c shows the measured 4-probe resistance of three different microcavities in the regime as a function of the carrier density above Gate 1 and Gate 2. Resistance peaks are visible near $|n_1| = |n_2|$ (red diamond), $|n_1| = 4|n_2|$ (green square), $4|n_1| = |n_2|$ (blue circle). At $|n_1/n_2| = 1/l^2$ or $l^2$ for integers $l$, a charge carrier can undergo two Veselago refractions and $l-1$ reflections at the charge neutrality boundary (Fig. 2d–h), forming a closed-loop trajectory back to its original position (see Supplementary Note 6 and 8) interfering with itself at the first cavity wall where it was originally injected. When the charge eventually leaves the cavity, the Veselago interference results in a higher chance of the electron escaping via the first cavity wall (reflected back to the source, and does not contribute to the current to the drain) than the opposite cavity wall (toward the drain, contributing to an uncollimated current to drain). In Fig. 2d, the bold arrow marks the main inference loop of the first-order interference peak as a consequence of two consecutive Veselago refractions. The faded line marks the trajectory of a smaller portion of carriers that are reflected (instead of refracted) at the pn-junction boundary in the center of the cavity, eventually contributing to the same resistance peak via two consecutive Veselago refractions, to the side of the main loop (see

Supplementary Note 10). Similarly, a closed interference loop (bold) can be formed at $|n_1| = 4|n_2|$ and $4|n_1| = |n_2|$, via a sequence of two refractions (at charge neutrality line close to the cavity boundary with lower doping) and three reflections, leading to the second-order Veselago interference peaks (see Supplementary Note 6 and 8). The Veselago interferences are reproduced in multiple devices (Fig. 2), with slight variations of microscopic details from realistic device specifics and inhomogeneity (see Supplementary Note 4, 7 and 9).

## Magnetic field, bias and temperature dependence

To further confirm the nature of Veselago interference, we measure the 4-probe resistance of the cavity under an out-of-plane external magnetic field. The magnetic field dependence shows gradual broadening and suppression of observed Vesalago interference peak (Fig. 3b–d), qualitatively agreeing with the expectation and simulated carrier trajectory (Fig. 3a and Supplementary Note 15), and are reproduced in multiple devices (see Supplementary Note 5) with realistic device variations in quantitative details. With a small magnetic field, the Lorentz force distorts the shape of interference loop for carriers that manage to reach the boundary at small incident angle (Fig. 3a). The reflection (refraction) of carriers at the pn-junction boundary is consequently enhanced (suppressed), reducing the probability of forming a close Veselago interference loop (see Supplementary Note 15). Figure 3e shows the measured resistance as a function of magnetic field and carrier density around the peak position at $|n_1| = |n_2| = 7 \times 10^{11}\,\mathrm{cm}^{-2}$, where $\Delta n = 0$ corresponds to the resonant condition. The width of the interference peak monotonically increases with magnetic field until it completely disappears. Due to the symmetry of the interference loop at $|n_1| = |n_2|$, no bias dependence is observed for the first-order peak (Fig. 3f). No significant temperature dependence is observed from $T = 4\,\mathrm{K} \sim 12\,\mathrm{K}$, as the specifics of the interference loop are expected to be insensitive to thermal excitations (Fig. 3g) (see Supplementary Note 12).

## Improving collimation via Veselago interference

The angle dependence of the Klein tunneling probability has been utilized to collimate electron flow in ballistic graphene devices[9,18]. However, carriers with small-incident angle can still pass through a series of Klein barriers with a non-negligible probability. This has been

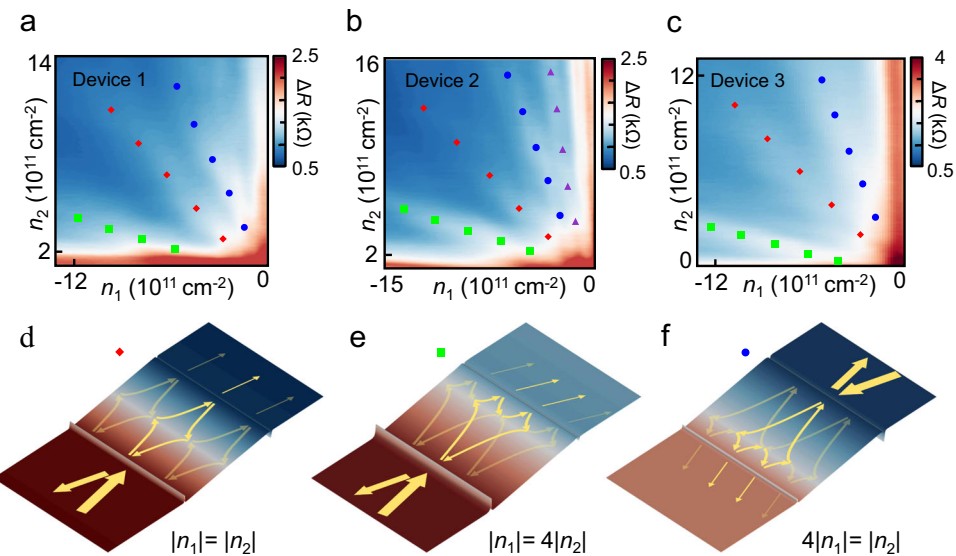

**Fig. 2 | Veselago interference and localization. a–c** Measured 4-probe resistance of cavity as a function of $n_1$ and $n_2$ in three devices. Resistance peaks are visible at $|n_1| = |n_2|$ (red), $|n_1| = 4|n_2|$ (green), $4|n_1| = |n_2|$ (blue) and $9|n_1| = |n_2|$ (purple), **d** When $|n_1| = |n_2|$, a symmetric pn-junction is defined in the cavity. Carriers injected into the cavity (with a low probability $p$ and a small incident angle) undergo Veselago

refraction, followed by reflection at the oppositive cavity boundary, then a second Veselago refraction to the original position. At $B = 0$, the process localizes carriers and gives rise to measured resistance peaks. **e, f** Similarly, a closed interference loop (bold) can be formed via a sequence of two refractions and three reflections at $|n_1| = 4|n_2|$ and $4|n_1| = |n_2|$, leading to a second-order Veselago interference peak.

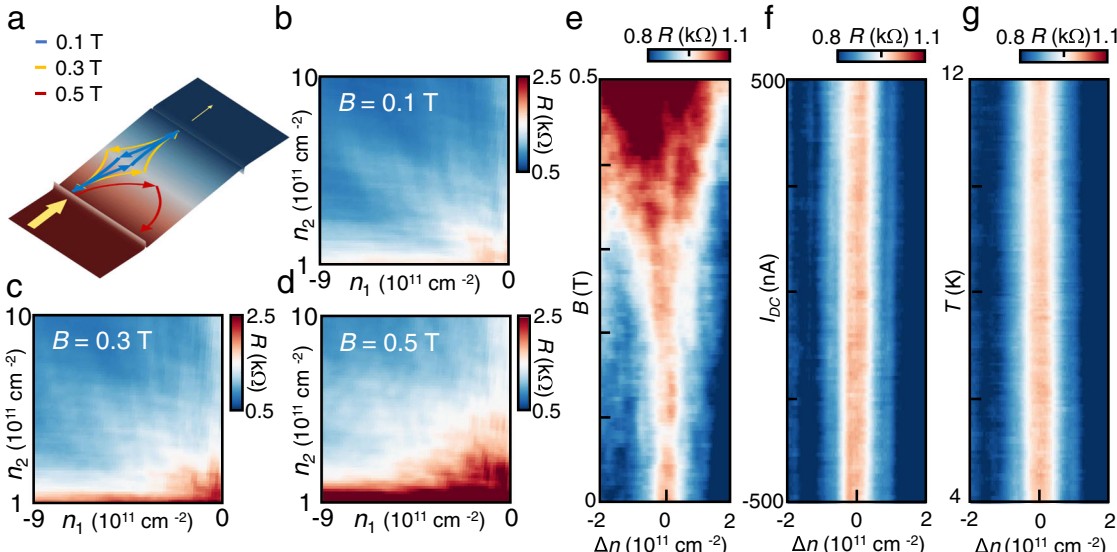

**Fig. 3 | Dependence on magnetic field, bias and temperature. a** Simulated typical carrier trajectories under various small magnetic fields. **b**, **c** Veselago interference peaks gradually broadens with increasing magnetic field and **d** is eventually suppressed when majority of the carriers fail to form the interference loop. **e** Measured resistance as a function of magnetic field shows the gradual broadening. **f** No bias dependence is observed for the first-order peak at $|n_1| = |n_2|$ due to the symmetry of the interference loop. **g** No significant temperature dependence is observed from $T = 4\,K - 12\,K$.

a major challenge for further improving collimation efficiency for advanced electron optics. Here, we show that Veselago interference can be used to further collimate carriers, particularly those with small incident angle. Figure 4a shows the device diagram for characterizing the collimation efficiency of Veselago interference. Source (drain) contacts are intentionally placed at the bottom right (left) corners of graphene on top of Gate 2 (1). Two 1 μm-wide voltage probes are placed at the top and bottom edges of the graphene on top of Gate 1. Carriers are injected from the source contact and collimated by the cavity wall, allowing only a small portion of carriers with small incident angle to enter the cavity. When $|n_1| < |n_2|$, the uncollimated carriers that escape the cavity predominantly reach voltage probe B, resulting in a transverse voltage proportional to the current density from uncollimated charge carriers (Fig. 4b). At $4|n_1| = |n_2|$, Veselago interference further localizes uncollimated carriers in the cavity (Fig. 4b), resulting in a near-zero measured transverse voltage amongst a high resistance background where Veselago interference is off-resonance, demonstrating its efficiency in further collimating small-incident-angle carriers that would otherwise manage to get across the cavity. The sharp drop in the transverse voltage at (and only at) the $4|n_1| = |n_2|$ (Fig. 4d) demonstrates the added collimation effect from the 2nd order Veselago interference, compared to general $|n_1| < |n_2|$ cases when Veselago interference and its collimation effect are absent. For the other two interference peaks at $|n_1| = 4|n_2|$ and $|n_1| = |n_2|$, the collimation efficiency is not characterized by the transverse voltage (Fig. 4c), as the ballistic carriers (collimated or not) will reach and be diffusively scattered from the physical edges of the device before reaching either voltage probe (see Supplementary Note 11).

A weak bias dependence (Fig. 4e) is observed for the zero transverse-resistance dip, as expected from the bias dependence of second-order peaks at $|n_1| = 4|n_2|$ (see Supplementary Note 9). As a perpendicular magnetic field is applied beyond $B = 0.3\,T$, the transverse voltage increases both due to the destruction of Veselago interference as well as the curved trajectory of carriers after passing through the cavity (Fig. 4f). No significant temperature dependence is recorded from $T = 4\,K - 12\,K$ (Fig. 4g), also consistent with our previous observations.

In conclusion, we have developed a novel device architecture for a microcavity in monolayer graphene with strain and electrostatic-field

engineering. We report a new electro-optics phenomenon: Veselago interference as a result of electron localization after two consecutive Veselago refractions in the cavity. The observed low-temperature magneto-transport signature agrees quantitatively with Veselago physics and provides further experimental evidence and insights into relativistic single-particle dynamics. Finally, the electrons that participate in Veselago interference in the cavity are those with small-incident-angle momenta, the exact same electrons primarily responsible for the relatively low collimation efficiency in previously studied pn-junction Klein collimators. By characterizing the collimation efficiency with transverse voltage probes, we provide proof-of-principle demonstration that Veselago interference helps to further localize these uncollimated electrons and thus improves the collimation of current through the cavity. Our work provides an important new technical and conceptual component toward advanced electron-optic circuits based on the precise manipulation of ballistic electron trajectories.

## Methods

### Gate fabrication

To form a strong strain at the boundary of the pn junction, the gate responsible for the p- and n-doped regions topographically differs in height. To achieve this, two main methods are implemented for the devices studied in this work. The first method (Device 3) employs two separate metal gates, each fabricated with a round of electron-beam lithography and evaporation (of different thickness), or alternatively one electron-beam lithography round while evaporating with a deliberate tilt of the wafer. In the second method (Devices 1 and 2), one gate is a metal bottom gate and the other is a silicon back gate (shown in Fig. S1). For each method, Cr/Pd-Au alloy (1 nm/varying thickness or 1 nm/7 nm) is deposited. The Pd-Au alloy (40% Pd/60% Au) is chosen to reduce surface roughness compared to conventional Au deposition. After liftoff, the gates are annealed in a high-vacuum environment at 350 °C for 5 minutes to remove resist residue and ensure atomically-flat topography.

### Device fabrication

We sequentially pick up hBN, graphene, and hBN flakes using polypropylene carbonate (PPC) and polydimethylsiloxane (PDMS) stamps via the standard dry transfer technique. The hBN-encapsulated

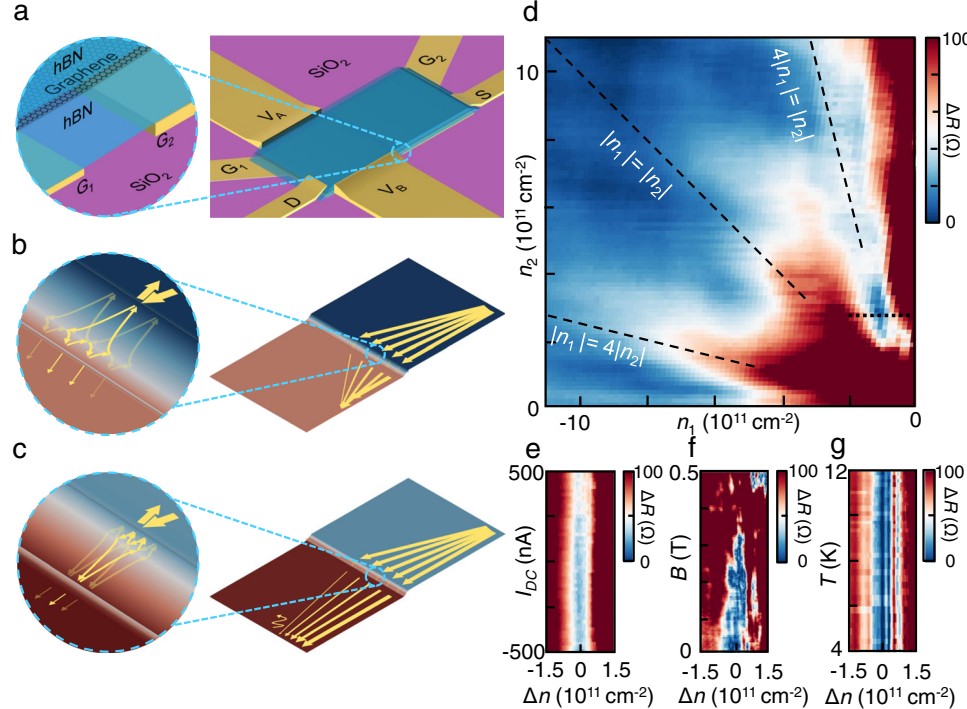

**Fig. 4 | Enhancing collimation with second-order Veselago interference.**
**a** Device diagram demonstrating the collimation of small-incident-angle electrons with Veselago interference. Carriers are injected from the source contact and first collimated (ineffectively) by the cavity wall. **b** When $|n_1| < |n_2|$, the uncollimated carriers that escape the cavity predominantly reach voltage probe B, while Veselago interference further localizes uncollimated carriers in the cavity at $4|n_1| = |n_2|$. This leads to **d** a near-zero measured transverse voltage at $4|n_1| = |n_2|$ in the midst of high resistance background. **c** When $|n_1| \geq |n_2|$, the collimation efficiency is not characterized by transverse voltages. **e** Weak bias dependence is observed for the zero transverse resistance dip. **f** As a perpendicular magnetic field is applied beyond 0.3 T, the measured transverse voltage increases both due to the destruction of Veselago interference. **g** No significant temperature dependence is recorded from $T = 4\,K$ - $12\,K$.

monolayer graphene stack is transferred on top of the prepatterned bottom gates. The sample is rinsed in acetone and IPA and annealed again in a high-vacuum environment at 350 °C for 5 minutes. The device regions on top of both gates as well as in the gap are atomically flat and strain-free after annealing, while lattice distortions are only introduced at the boundaries of two bottom gates. Electrical contacts to gates and ohmic contacts to 1D boundaries of the graphene stack are made by electron-beam lithography, dry-etching and subsequent metal deposition (Cr/Pd/Au, 1 nm/5 nm/140 nm). A final round of dry-etching with an electron-beam lithography etch mask defines the lateral geometry of the devices.

## Data availability

The data generated in this study have been deposited in the GitHub public repository without accession code: X1Zhang123/Source-Data-Gate-tunable-Veselago-Interference-in-a-Bipolar-Graphene-Microcavity-(github.com).

## Code availability

Relevant code of carrier trajectory simulations are provided with this paper and can be found at X1Zhang123/Source-Data-Gate-tunable-Veselago-Interference-in-a-Bipolar-Graphene-Microcavity- (github.com). All other code are available from the corresponding author upon reasonable request.

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

## Acknowledgements
We thank Boris Shklovskii and Alex Kamenev for useful discussions. The work at UMN was supported by the National Science Foundation CAREER Award NSF-1944498. Portions of the UMN work were conducted in the Minnesota Nano Center, which is supported by the National Science Foundation through the National Nano Coordinated Infrastructure Network (NNCI) under Award Number ECCS-1542202. The band structure calculation by Z.Z., M.L. and E.K. is supported by STC Center for Integrated Quantum Materials, NSF Grant No. DMR-1231319, ARO MURI Grant No. W911NF14-0247, and NSF DMREF Grant No. 1922165. Calculations were performed on the Odyssey cluster supported by the FAS Division of Science, Research Computing Group at Harvard University and the National Energy Research Scientific Computing Center (NERSC), a U.S. Department of Energy Office of Science User Facility located at Lawrence Berkeley National Laboratory, operated under Contract No. DE-AC02-05CH11231. Portions of the hexagonal boron nitride material used in this work were provided by K.W. and T.T. K.W. and T.T. acknowledge support from the Elemental Strategy Initiative conducted by the MEXT, Japan (Grant Number JPMXP0112101001) and JSPS KAKENHI (Grant Numbers 19H05790, 20H00354 and 21H05233).

## Author contributions
K.W. designed the experiment and device architecture. Data presented in this work were taken by X.Z. and W.R. Device fabrication, measurement and data analysis was performed by X.Z. and W.R. with support from E.B., K.T. and Y.L. under the supervision of K.W. The band structure calculation for strained graphene was performed by Z.Z. under the supervision of E.K. and M.L. A portion of the hexagonal boron nitride used in this work was provided by T.T. and K.W. X.Z., W.R., E.B. and K.W. wrote the paper. All authors discussed the results and provided comments on the manuscript.

## Competing interests
The authors declare no competing interests.
