## [Peer Review File · Nature Communications]

REVIEWER COMMENTS

Reviewer #1 (Remarks to the Author):

The authors report a novel device architecture based on local strain and PN junction induced by uneven split gates geometry. The local strain along the edges of the gates acts as barrier to form a micro-cavity and collimates carriers from Klein tunneling. The PN junction inside the cavity acts as a Veselago lens to bring the carrier back to its original position and constructively interfere with itself. The temperature, magnetic-field, and bias dependences on the interference peaks are studied.

Although the resistance peaks might have an interesting origin, I find that the model the authors give does not explain the results. In addition, the model also relies on strain induced band gap which have been shown to be extremely hard as it requires very large strain near the breaking point of graphene (see ref. 27).

I have the following comments and suggestions

1) The authors claim that the carriers travel inside the cavity and leave will interfere constructively with "itself" at the first barrier. However, it seems to me that the constructive interference can't always be satisfied. The carriers travel inside the cavity will pick up a phase of kL where k is the wave vector and L is the travelling distance inside the cavity (assuming that there is no phase shift from the reflection at the boundary). So, to interfere constructively, the value of kL has to be an integer multiple of 2π . However, the carries with different incidence angles will travel at different distance L . Therefore, the values of kL can't always be 2π . In addition, the wave vector varies with the density. So, along $|n_1| = |n_2|$ line, the value of k will vary, and interference fringes should be observed which is not the case in the data.

2) If the observed peaks are the result of quantum interference, it should depend sensitively on temperature due to the decoherence. However, the data show very weak temperature dependence of the peaks. (see Ref 8. fig. 3c in which the oscillation peaks reduce significantly already at 16 K).

3) The authors use the magnetic length as a length scale for the trajectory of the carriers. However, the correct length scale to describe the trajectory under magnetic field should be cyclotron radius (Radius = $\hbar v_F k_F / eB$) which depends on both magnetic field and carrier density.

4) The mobility of the devices is claimed to be $\sim 300,000 \text{ cm}^2/\text{V}\cdot\text{s}$. How do the authors obtain this number? What is the field effect mobility? Given such a high mobility, I would expect to observe Shubnikov de Haas oscillations already at 0.5 T in the magnetic-field dependent data. However, no oscillation is observed.

5) The authors claim that the lattice distortion is “apparent” in the AFM measurement in Fig. 1c. Given the spatial resolution of the AFM and the data itself, it’s not clear if there is any lattice distortion. The graphene can smoothly bend down to the lower gate since it’s supported by a much thicker BN. I find it hard to believe that an 8 nm step over 50 nm separation will induce enough strain to change the band structure of graphene.

6) Do the authors have any other evidence that they can create significant strain on graphene with the uneven split gate geometry? For example, do you observe any shift in 2D or G bands in Raman spectrum?

7) If the authors can indeed induce such large strain, how can they be sure that Klein tunneling can still be applied? Klein tunneling is a property of massless Dirac fermion. However, with large strain, the band structure is no longer linear as shown in the DFT calculation. Therefore, the transmission probability will also be different.

8) The optical image of the device in Figure 1b is not very clear. It’s hard to tell which are BN/graphene/BN, bottom gates, and contacts. What’s the size of the scale bar?

9) What is the pseudo-gap created by an electrostatically-defined pn-junction mentioned in page 2?

10) In figure 3d, e, and f, what are the values of n_1 and n_2 ? Do the B , I_{DC} , and T dependence the same for other $|n_1| = |n_2|$ positions.

Reviewer #2 (Remarks to the Author):

An experimental search for Veselago lensing in graphene heterojunctions has been going on for more than a decade. The manuscript "Gate-tunable Veselago Interference in a Bipolar Graphene Microcavity" reports the first experimental demonstration of the phenomenon, a demonstration which is in my opinion convincing, elegant, and potentially consequential for applications not directly related to Veselago lensing itself, such as the collimation of electron flows in graphene.

The experiment is cleverly designed, combining a strain-induced Fabry-Perot interferometer with negative-angle refraction at a graphene n-p junction in such a way as to make Veselago lensing leave a robust signature on such a simple observable as the [four-terminal] resistance of the sample. Several independent cross-checks leave little doubt as to the mechanism causing the observed resistance peaks. As a bonus, the authors demonstrate how their experimental design can be used to enhance the collimation of electric currents as compared with a simple n-p junction.

The experiment is top notch and Nature Communications would undoubtedly benefit from publishing this work.

The manuscript is well written and easy to follow.

I only have a couple of questions, which I am sure the authors will not find difficult at all to clarify

1. Most of the experimental data are presented as something

being a function of n_1 , n_2 or Δn . What method is used to measure the actual carrier densities?

2. The hypothetical trajectories of electrons in Fig 1a and other figures are slightly bent. Have they been simulated or are they just an artist's impression of how a pseudo-relativistic electron should move in an inhomogeneous potential background? If the bending effect is as significant as is seen in the figures, should it not make higher-order resonant resistance peaks, e.g. $n_1 = -4 n_2$ significantly broader than the main peak $n_1 = -n_2$?

Reviewer #3 (Remarks to the Author):

Authors studied bilayer graphene microcavity formed by split local gates and strained graphene. They measured resistance of the device with varying gate voltages, and showed resistance peak features at specific conditions of carrier density of two regions of graphene. They claim this is due to 'Veselago interference' and this can further enhance the electron collimation through p-n junctions of graphene.

Monolayer graphene is a unique platform to study electronic optics based on relativistic Dirac fermions. The actual collimation efficiency has been quite low in p-n junction, in contrast to the theoretical expectation. Authors suggest consecutive Veselago refraction in p-n junction can happen due to potential barriers formed by strain, and this can reject uncollimated electron flow back to the source. This is indeed an interesting and novel idea for enhancing collimation efficiency, and potentially has an impact on electronics optics studies in graphene.

However, there are a few things to be clarified before deciding the recommendation for publication.

1. According to the reference 14 in the supplementary information, the gap opens only when the strain direction is along zigzag. How do we know this is the case for the experiment in the manuscript? Is there any signature in transport of gap opening at the boundary of the bottom local gate?

2. In the sentence "The absence of these transport features in the nn or pp gate configurations rule out the possibility for them to arise from disorder or Fabry-Pérot interference.", authors claim that resistance peak appearing only in bipolar gate configurations cannot be due to F-P interference. However, the formation of p-n junction can play a crucial role of giving F-P interference as they can act as a reflector. Indeed, graphene p-n-p or n-p-n junctions routinely show F-P interference. I wonder how F-P interference can be ruled out for giving resistance peaks.

3. In the sentence "Due to the symmetry of the interference loop at $|n_1|=|n_2|$, no bias dependence is observed for the first-order peak (fig. 3e).", how about the bias dependence for 2nd order peaks ($4|n_1|=|n_2|$ or $|n_1|=4|n_2|$)?

4. I failed to follow the logic of the discussion on Fig. 4.

- In the sentence "When $|n_1| < |n_2|$, the uncollimated carriers that escape the cavity predominantly reach voltage probe B, resulting in a transverse voltage proportional to the current density from uncollimated charge carriers (fig. 4b).", authors discuss the situation for $|n_1| < |n_2|$. However, left circular shape of panel in fig. 4b corresponds to the case for $4|n_1|=|n_2|$ which is the same with Fig. 2f.

- Authors say transverse voltage would show some voltage for $|n_1| < |n_2|$, but almost zero for $4|n_1|=|n_2|$. However, the condition " $|n_1| < |n_2|$ " includes the condition " $4|n_1|=|n_2|$ ". How can these two conditions give different results?

- I cannot understand the sentence "For the other two interference peaks at $|n_1|=4|n_2|$ and $|n_1|=|n_2|$, the collimation efficiency is not characterized by the transverse voltage (fig. 4c), as the ballistic carriers (collimated or not) will reach and diffusively scatter from the physical edges of the device before reaching either voltage probe (see SI for more details)." Which part of IS do authors refer to?

- Why is the outgoing direction of electrons is slightly tilted downward in the left panel of Fig. 4b, while outgoing direction of electrons is perpendicular to the p-n junction barrier in the left panel of Fig. 4c? Shouldn't $|n_1|=4|n_2|$ and $4|n_1|=|n_2|$ be symmetric?

Minor comments:

1. Many sentences of the figure captions are repeated in the main text. Please remove redundancy.
2. Authors should specify which part of SI is referred at every point they are referred.
3. How was the mobility $\sim 300,000 \text{ cm}^2/\text{Vs}$ estimated?

Responses to Referees' Comments

Reviewer Comments: blue; Responses: Black; Changes Made in Revision: Red; Deletion: Gray
Referee 1

The authors report a novel device architecture based on local strain and PN junction induced by uneven split gates geometry. The local strain along the edges of the gates acts as barrier to form a micro-cavity and collimates carriers from Klein tunneling. The PN junction inside the cavity acts as a Veselago lens to bring the carrier back to its original position and constructively interfere with itself. The temperature, magnetic-field, and bias dependences on the interference peaks are studied. Although the resistance peaks might have an interesting origin, I find that the model the authors give does not explain the results. In addition, the model also relies on strain induced band gap which have been shown to be extremely hard as it requires very large strain near the breaking point of graphene (see ref. 27).

We thank referee's time in reading our manuscript. We also thank the referee for his/her comments on manuscript and raising constructive and helpful suggestions for improving the paper's overall presentation accuracy and consistency. We have carefully addressed each of the comments from the referee, listed point by point below.

I have the following comments and suggestions

1) The authors claim that the carriers travel inside the cavity and leave will interfere constructively with "itself" at the first barrier. However, it seems to me that the constructive interference can't always be satisfied. The carriers travel insides the cavity will pick up a phase of kL where k is the wave vector and L is the travelling distance inside the cavity (assuming that there is no phase shift from the reflection at the boundary). So, to interfere constructively, the value of kL has to be an integer multiple of 2π . However, the carries with different incidence angles will travel at different distance L . Therefore, the values of kL can't always be 2π . In addition, the wave vector varies with the density. So, along $|n_1| = |n_2|$ line, the value of k will vary, and interference fringes should be observed which is not the case in the data.

Referee 1 raised a series of insightful questions about our results following the consideration of spatial phase kL . The method has been widely used for interpreting Fabry-Pérot (F-P) interference where the carrier density (and therefore k) and incident angle (and therefore L) are well-defined. However, we want to emphasize that several aspects of the physical process studied in this work are distinct from conventional F-P interference, particularly in terms of the implications of the spatial kL phase, specifically:

Any given data point on the resistance peak is contributed by many close-loop electron trajectories with different incident angles, each with their own kL phases. This does lead to a resistance peak at resonant condition (when closed-loop path is formed for all incident angles), but we do not expect kL phase coherence nor oscillatory dependence on k . The Klein tunneling into the cavity resulted from a distribution of interference loops, each corresponding to carriers injected at a different incident angle (fig. R1). For any given data point, the measured resistance is the summation of resistance peaks from all the interference loops that are simultaneously contributing. A

Figure R1. Interference Paths in the Microcavity. (a) We pick five Veselago interference (from many) paths with different incident angles, labeled by blue, green, yellow, orange and red color, respectively. The dominant contribution to Veselago interference is indicated as the orange path. Different incident angles can bring significant changes in the interference paths and kL phases. (b) Illustration of interference amplitude corresponding to each path in (a) as a function of incident k with distinct oscillation frequency. (c) Expected outcome of summing over interference amplitudes from all paths (not limited to this five) as a function of incident wave vector k . The k dependence is exponential decaying instead of oscillatory behavior.

simple qualitative demonstration for the implication of co-existing interference loops can be shown below:

As a result, the dependence on carrier density (or starting k when an electron is injected into the cavity) should be a peak with exponentially decaying peak height (as a function of k) instead of fringes. To help visualize this point, figure R1b plots out the interference fringes from each contributing closed loop (same color in fig. R1a) which does show oscillatory behavior, and their summation (fig. R1c) which shows a peak height that decays with k , consistent with our observation.

To emphasize this point, we have added discussion in the manuscript “**Any given data point on the resistance peak is contributed by many closed-loop carrier trajectories with different incident angles, each with its own kL phase, where k is the wave vector and L is the carrier travelling distance. While this does lead to a resistance peak at resonant condition (when closed-loop paths are formed for all incident angles), the dependence of peak height on k is monotonic instead of oscillatory (see SI S12)**” and a new SI section S12 to demonstrate more detailed and quantitative description of the effect has been added.

2) If the observed peaks are the result of quantum interference, it should depend sensitively on temperature due to the decoherence. However, the data show very weak temperature dependence of the peaks. (see Ref 8. fig. 3c in which the oscillation peaks are reduced significantly already at 16 K).

Following our model for the previous question: decoherence mechanisms such as temperature disrupt the coherent oscillatory behavior for any given interference loop in the cavity. However,

many interference loops co-exist in the cavity each with their own oscillatory frequency (as a function of k) depending on their specific incident angles. The phase-coherent signature of any given interference loop, even in the absence of decoherence, is not represented in our data as interference fringes to begin with, but instead as a smooth decay function of k as a collective behavior of all interference loops that contribute to the measured resistance simultaneously. Therefore, our data is expected to be fairly unaffected by the implication of phase-decoherence, until such effect is large enough to “broaden” even the exponential decay profile (fig. R1c) at high temperature.

3) The authors use the magnetic length as a length scale for the trajectory of the carriers. However, the correct length scale to describe the trajectory under magnetic field should be cyclotron radius (Radius = $\hbar v_F k_F / eB$) which depends on both magnetic field and carrier density.

We thank the referee for carefully reading our paper, and for pointing out this important point. We agree with the referee that the cyclotron motion is the essential physics process here. In our related figures in the manuscript and SI S15, we in fact plotted the cyclotron orbits as the mechanism responsible for magnetic field dependence. However, in the SI S15, we accidentally used the magnetic length for the discussion (which was meant for a comparing magnetic confinement with the cavity confinement), which is irrelevant to the physics picture we presented in our figures. We have corrected the corresponding discussion and SI section S15.

4) The mobility of the devices is claimed to be $\sim 300,000 \text{ cm}^2/\text{V}\cdot\text{s}$. How do the authors obtain this number? What is the field effect mobility? Given such a high mobility, I would expect to observe Shubnikov de Haas oscillations already at 0.5 T in the magnetic-field dependent data. However, no oscillation is observed.

We thank the referee for raising this important question. The mobility $\sim 300,000 \text{ cm}^2/\text{V}\cdot\text{s}$ is a rough estimation from observing half-width of the Dirac peak, compared to that of a typical Hall bar device with known mobility. We believed that the accurate determination of device mobility is not critical for the physics phenomena described in this work, for as long as the mean-free path is

Figure R2. Fitting of Resistance Versus Carrier Density. Blue curve is the original data of resistance as a function of carrier density, while the red curve is the fitting result. The mobility is characterized about $90,000 \text{ cm}^2 \text{ V}^{-1} \text{ s}^{-1}$.

sufficiently longer than the width of the cavity to ensure ballistic nature of electron within the interference loop. Unlike a homogeneous Hall bar device, our device architecture is inhomogeneous by design with the presence of strain-defined cavity boundaries. A more quantitative estimation of the carrier mobility can therefore be less straightforward when using similar methods for Hall bar devices. However, we still performed new experimental characterizations following the referee’s suggestions, and provided some elaboration on how to interpret the results in the context of the more complicated device architecture.

Method 1: Estimation of carrier mobility using 4-probe resistance at zero magnetic field. Unlike a Hall bar device, the cavity

boundaries add significant resistance to the 4-probe R_{xx} at low field, making estimation of mobility using R_{xx} an underestimation of graphene quality. However, for a lower-bound estimation of mobility, we can still fit the Dirac peak (when both regions are p doped, in the absence of Veselago interference) by using the following formula¹

$$R = R_c + \frac{L}{W} \frac{1}{\mu e \sqrt{n_0^2 + (C_g |V_g - V_{\text{Dirac}}|)^2}}$$

where L and W are the length and width of the sample, e is the charge magnitude of an electron, R_c is the contact resistance from the graphene 1D contact, μ is the mobility, n_0 is the residual carrier density, C_g is the capacitance of the hBN per unit area, V_g is the gate voltage, and V_{Dirac} is the Dirac peak shift voltage. The mobility is estimated as $\sim 90,000 \text{ cm}^2 \text{ V}^{-1} \text{ s}^{-1}$. Note that the fitting formula assumes a single uniform mobility (and conductivity) for the entire device, while multiple high-resistance and low-mobility strained boundaries significantly contribute to the data being fitted. Therefore, we expect this to be a lower-bound estimation, with the actual device mobility significantly higher and closer to our previous rough estimation ($\sim 300,000 \text{ cm}^2 \text{ V}^{-1} \text{ s}^{-1}$). However, the lower bound estimation of the mean-free path² of $0.9 \text{ }\mu\text{m}$, which is one order of magnitude larger than the cavity width, is sufficiently large to ensure uninterrupted interferences loops in the cavity.

Method 2: Using SdH oscillations or quantum Hall effect to estimate the carrier mobility.

Unlike Hall bar devices, the cavity boundary also prevents quantum Hall transport as the edge states becomes unequilibriumed at the cavity boundary³, making it difficult to accurately extract the mobility from SdH oscillations. This is why SdH oscillations are not observed in the micro-cavity device: even when the different regions of the sample are doped with the same carrier density, the locally-strained boundary between gates can prevent the effective transmission of the quantum Hall edge states, and the transmission rate reduces as the magnetic field increases the effective gap at the strained boundary ($\Delta = \Delta_0 + \hbar\omega$). We have measured a different device region

Figure R3. Quantum Hall Edge States in pn and pp Junctions. (a)-(b) Locally-strained boundary can prevent the effective transmission of the quantum Hall edge states. (c) SdH oscillation can be observed starting at $B = 1 \text{ T}$ in a different device from the same stack.

fabricated with exactly the same graphene stack (as the cavity studied in this work), but without strain and gate-defined boundaries. The SdH oscillations start to be visible at ~ 1 T, giving an estimation of carrier mobility lower-bound of $90,000 \text{ cm}^2 \text{ V}^{-1} \text{ s}^{-1}$, or a mean-free path lower-bound⁴ of $0.9 \text{ }\mu\text{m}$ with the $1.5 \times 10^{12} \text{ cm}^{-2}$ carrier density that is an order of magnitude longer than that of interference loop. We have added a new SI section S4 “Lower-bound Estimation of Carrier Mobility” to emphasize these important technical details.

5) The authors claim that the lattice distortion is “apparent” in the AFM measurement in Fig. 1c. Given the spatial resolution of the AFM and the data itself, it’s not clear if there is any lattice distortion. The graphene can smoothly bend down to the lower gate since it’s supported by a much thicker BN. I find it hard to believe that an 8 nm step over 50 nm separation will induce enough strain to change the band structure of graphene.

We thank the referee for raising this important technical point. For a Klein barrier to be effectively defined with angle-dependence, a smooth distortion is all we need (and in fact preferred over a sharp atomic distortion). The AFM topography measurement in Fig. 1c shows that the graphene is smoothly bended down for an 8 nm step over 50 nm separation. We can see how this becomes a confusion raised by our way of presentation. The theoretical calculation⁵ shows the band structure of strain over a few unit cells, simply because a quantitative calculation for the experimental reality^{6,7}, a smooth strain over 8 nm, is unrealistic for our theory collaborators. The theoretical band calculation is intended to qualitatively show that atomic strain can results into bandgap. It is not intended to show that the device needs a sharp atomic distortion (over a few unit cells) or a gap as large as 0.4 eV to operate as it designed to be. In fact, a large strain-induced gap will result into an extremely sharp angle-dependence of Klein tunneling rate, rendering effective no carriers with finite incident angle to enter the cavity, let alone Veselago inference. A smooth atomic strain is not only capable of hosting Veselago interference, but actually preferred and an integral part of the experimental design.

Besides, a quantitative calculation of the band structure with realistic smooth strain is unnecessary, for two reasons: **1. Device variation in local atomic strain is expected. But device functionality and Veselago interference does not require a precise strain condition.** As long as the strain remains smooth enough, the variation of effective Klein barrier height does not change the fact that the barrier allows finite-angle-incident carriers to enter the cavity with low (but non-zero) Klein tunneling probabilities. Therefore, the core device functionality and the observation of Veselago interference is not sensitive to the particular strength of atomic strain, as demonstrated by reproductions of core results in multiple samples. This is in fact is a significant advantage for its device application in future electron-optics. **2. We directly measure the resistance of the strain-induced barrier and compare it to previously-studied Klein barriers.** When biased at pp or nn region (in absence of pn junctions), the 4-probe resistance of the device can be written as $R_{\text{tot}} = R_G + 2R_{\text{cw}}$, where R_G is the graphene resistance, R_{cw} is the resistance from a single cavity wall. At high doping (when $n_1 \sim n_2 \sim 10^{12} \text{ cm}^{-2}$), R_{tot} of the cavity device is measured to be $\sim 0.32 \text{ k}\Omega$. At the same doping, R_G is estimated to be $\sim 0.01 \text{ k}\Omega$, from measuring a different region of the same graphene stack without any strain-induced or electrostatically defined barrier. This gives an estimation of $R_{\text{CW}} \sim 0.155 \text{ k}\Omega$, similar to that of a Klein barrier defined by a pn-junction with known ineffective collimation (or finite Klein angle-dependence) preferred by our device architecture. We have added a new SI section S14 to elaborate these technical points.

To improve the clarity of main manuscript, we have (1) moved the theoretical band calculation to the SI and emphasized that the calculation was meant for the qualitative

demonstration that a gap can be opened by strain. The decision to perform the calculation over few unit cells is because of the limitation of the computational effort needed for a scenario more similar to the experimental parameters (smooth bend). (2) Added to the main narrative “A smooth strain over 8 nm is apparent in the AFM topography, resulting into an effective Klein barrier of net resistance 0.155 k Ω (See SI S14), similar to that of previously-reported electrostatically-defined Klein barriers⁸. While the Klein-tunneling angle dependence allows rough collimation of injected current, electrons with finite-incident angle can tunnel through the strain-defined barrier with low injection rate, making the regions in between two barriers an electronic analogy of an optical cavity”.

6) Do the authors have any other evidence that they can create significant strain on graphene with the uneven split gate geometry? For example, do you observe any shift in 2D or G bands in Raman spectrum?

As we elaborated in the previous response, we do not intend to create significant strain on the graphene. We apologize again for the confusion arose from our presentation of theoretical calculation on the case as it does. We thank the referee for raising this important technical question.

7) If the authors can indeed induce such large strain, how can they be sure that Klein tunneling can still be applied? Klein tunneling is a property of massless Dirac fermion. However, with large strain, the band structure is no longer linear as shown in the DFT calculation. Therefore, the transmission probability will also be different.

As we annealed the device at 350 °C for 5 minutes, the strain is fully relaxed over the device region (cavity) that are between the two barriers. The band structure of the graphene in and out of cavity remains pristine with linear dispersion relationship, except for the barrier itself, where Klein tunneling applies. We have emphasized this important point in our narrative “We annealed the device at 350 °C for 5 minutes, the strain is fully relaxed for the device region (cavity) that are between the two barriers. The band structure of the graphene in and out of cavity remains pristine with linear dispersion relationship, expect for the barrier itself. This is an experimental design so that Klein tunneling and Veselago physics remain”.

8) The optical image of the device in Figure 1b is not very clear. It's hard to tell which are BN/graphene/BN, bottom gates, and contacts. What's the size of the scale bar?

We thank referee again for catching small details in our manuscript. The optical image is already at the highest resolution for the optical microscope in our lab. As the whole sample size is only few micros, several times of light wavelength, the edges of the device might become blurry. The scale bar represents 5 μm . We have updated this information in the caption.

9) What is the pseudo-gap created by an electrostatically-defined pn-junction mentioned in page 2?

We assume the “pseudo-gap” that the referee refers to is defined in this context of Klein tunneling angle dependence (or collimation efficiency). As previously mentioned, the measured resistance of the strain-induced barrier is similar to that of gate-defined pn junction. Therefore, for an rough order of magnitude estimation, the pseudo-gap is comparable to that of the previously-reported gate-defined pn junctions^{9,10} (~ 1 eV). In our device architecture, two strain-induced barriers co-exist and none can be turned on/off similar to that of electrostatically-defined pn junctions. Therefore, a more direct characterization of the collimation efficiency from a single strain-induced

barrier, or the electrostatically defined pn junction, cannot be performed in a similar quantitative way. Such characterization is not relevant to the central thesis of this work. Veselago interference is observed for as long as the Klein-tunneling angle-dependence allows electrons with finite incident angles to transmit (non-perfect collimation efficiency), which holds true for a wide range of electrostatic configuration (and pn junction height) as observed in our experiment. And the improved collimation efficiency demonstration is a result of Veselago interference, instead of Klein tunneling that facilitates it. The very emphasis of the data presented in fig. 4 is to demonstrate an improvement of collimation despite of a non-ideal collimation efficiency (or pseudo-barrier) that realistically can be established in the state-of-the-art experiments (defined electrostatically or via “smooth” strain).

10) In figure 3d, e, and f, what are the values of n_1 and n_2 ? Do the B , I_{DC} , and T dependence the same for other $|n_1| = |n_2|$ positions.

In figure 3d, e, and f, the value of both n_1 and n_2 are $7 \times 10^{11} \text{ cm}^{-2}$. To make it clearer, we add the carrier density information in the manuscript. We change “Figure 3d shows the measured resistance as a function of magnetic field and carrier density around the peak position, where $\Delta n = 0$ corresponds to the peak position” to “Figure 3d shows the measured resistance as a function of magnetic field and carrier density around the peak position at $n_1 = n_2 = 7 \times 10^{11} \text{ cm}^{-2}$, where $\Delta n = 0$ corresponds to the peak position”. We also took the data of magnetic field B , and T dependence at other $|n_1| = |n_2|$ positions for the first order interference peak. The measured dependences look similar as figure 3d, and f. We present the extra data as fig. R4a and R4b shows. For the dc Bias data, we didn’t measure extra data at different n_1 and n_2 carrier densities, but we did measure dc

Figure R4. Dependence on Magnetic Field, Bias and Temperature of Device 4. (a) Measured resistance as a function of magnetic field and carrier density around the peak position. (b) Weak temperature dependence can be observed from 4K to 12K for first-order peaks in device 1. (c) Bias dependence for first-order peaks at $|n_1|=|n_2|$ in device 4.

bias of the first-order peak in another device (device 4), which is shown as figure S5b in SI section S6. We put the image as fig. R4c here.

Referee 2

An experimental search for Veselago lensing in graphene heterojunctions has been going on for more than a decade. The manuscript "Gate-tunable Veselago Interference in a Bipolar Graphene Microcavity" reports the first experimental demonstration of the phenomenon, a demonstration which is in my opinion convincing, elegant, and potentially consequential for applications not directly related to Veselago lensing itself, such as the collimation of electron flows in graphene.

The experiment is cleverly designed, combining a strain-induced Fabry-Perot interferometer with negative-angle refraction at a graphene n-p junction in such a way as to make Veselago lensing leave a robust signature on such a simple observable as the [four-terminal] resistance of the sample. Several independent cross-checks leave little doubt as to the mechanism causing the observed resistance peaks. As a bonus, the authors demonstrate how their experimental design can be used to enhance the collimation of electric currents as compared with a simple n-p junction.

The experiment is top notch and Nature Communications would undoubtedly benefit from publishing this work. The manuscript is well written and easy to follow. I only have a couple of questions, which I am sure the authors will not find difficult at all to clarify

-Most of the experimental data are presented as something being a function of n_1 , n_2 or Δn . What method is used to measure the actual carrier densities?

We thank the referee for raising this important point. We follow the standard method in estimating the carrier density. The graphene and silicon back gate can be effectively treated as parallel-plate capacitor, in between which SiO_2 ($d_{\text{SiO}_2} = 285 \text{ nm}$, $\epsilon_{\text{SiO}_2} = 3.90$) and the bottom hBN ($d = 81.5 \text{ nm}$, $\epsilon_{\text{hBN}} = 3.76$) serve as dielectrics. The capacitance for graphene region on top of local metal gate is $c_{\text{gate}} = \frac{\epsilon_{\text{hBN}}\epsilon_0}{d} = 4.08 \times 10^{-8} \text{ C V}^{-1}\text{cm}^{-2}$. Similar method can be done for the capacitance between the silicon backgate and the graphene, therefore is $c_{\text{SiO}_2} = \frac{\epsilon_{\text{SiO}_2}\epsilon_0}{d_{\text{SiO}_2}} \cdot c_{\text{gate}} / \left(\frac{\epsilon_{\text{SiO}_2}\epsilon_0}{d_{\text{SiO}_2}} + c_{\text{gate}} \right) = 9.34 \times 10^{-9} \text{ C V}^{-1}\text{cm}^{-2}$. The carrier densities on the silicon backgate and local metal gate are $n_{\text{SiO}_2} = \frac{c_{\text{SiO}_2}V_{\text{SiO}_2}}{e}$ and $n_{\text{gate}} = \frac{c_{\text{gate}}V_{\text{gate}}}{e}$, respectively.

-The hypothetical trajectories of electrons in Fig 1a and other figures are slightly bent. Have they been simulated or are they just an artist's impression of how a pseudo-relativistic electron should move in an inhomogeneous potential background? If the bending effect is as significant as is seen in the figures, should it not make higher-order resonant resistance peaks, e.g. $n_1 = -4$ n_2 significantly broader than the main peak $n_1 = -n_2$?

We thank the referee for raising this important point. The trajectories depicted in figure 1a is a cartoon intended to qualitatively illustrate the essence of the physical process: (1) multiple closed loops (from electrons injected with different angles) co-exist (2) the loops are bent because the carrier density is spatially varying across the cavity (as the referee correctly pointed out). The resistance peak is observed when a closed loop can be formed, independent of the incident angle (or resulting size of the loop) and particular shape of the loop (bend versus straight), and therefore

we believe an illustration of the interference path is more appropriate here than a quantitative and elaborate simulation.

Regarding the width of the second order peak, we thank the referee for the careful reading of our paper and for raising this very insightful point. The broadening of the resonant resistance peaks is due to the spatial inhomogeneity of the cavity barriers (or roughness), which deviates from the assumption that they are “perfectly flat” when compared to the wavelength of the carriers reflected/refracted at these barriers. The 2nd order interference peaks are sharper because of the overall smaller carrier density at the boundary where carriers are reflected twice consecutively, as a larger carrier wavelength makes it less sensitive to the spatial roughness that results into the more significant broadening of the first order peak. We have further elaborated this point in SI section S9 “Width and DC Bias Dependence of 2nd order Veselago Interference Peaks”.

Referee 3

Authors studied bilayer graphene microcavity formed by split local gates and strained graphene. They measured resistance of the device with varying gate voltages, and showed resistance peak features at specific conditions of carrier density of two regions of graphene. They claim this is due to ‘Veselago interference’ and this can further enhance the electron collimation through p-n junctions of graphene.

Monolayer graphene is a unique platform to study electronic optics based on relativistic Dirac fermions. The actual collimation efficiency has been quite low in p-n junction, in contrast to the theoretical expectation. Authors suggest consecutive Veselago refraction in p-n junction can happen due to potential barriers formed by strain, and this can reject uncollimated electron flow back to the source. This is indeed an interesting and novel idea for enhancing collimation efficiency, and potentially has an impact on electronics optics studies in graphene.

However, there are a few things to be clarified before deciding the recommendation for publication. 1. According to the reference 14 in the supplementary information, the gap opens only when the strain direction is along zigzag. How do we know this is the case for the experiment in the manuscript? Is there any signature in transport of gap opening at the boundary of the bottom local gate?

We thank referee for raising this important technical point. The AFM topography measurement in Fig. 1c shows that graphene is smoothly bent down for 8 nm step over 50 nm separation. For a Klein barrier to be effectively defined with angle-dependence, a smooth distortion along any direction is all we need (and in fact preferred over sharp atomic distortion). The theoretical calculation⁵ shows the band structure of strain over a few unit cells, and along zigzag, simply because a quantitative calculation for the experimental reality^{6,7}, a smooth strain over 8 nm, is unrealistic for our theory collaborators. The theoretical band calculation is intended to qualitatively show that atomic strain can result into a bandgap. It is not intended to show that the device needs a sharp atomic distortion (over a few unit cells) along the zig zag direction or a gap as large as 0.4 eV to operate as it designed to be. In fact, a large strain-induced gap will result into an extremely sharp angle-dependence of Klein tunneling rate, effectively rendering no carriers with finite incident angle to enter the cavity, let alone Veselago inference. A smooth atomic strain is not only capable of hosting Veselago interference, but actually preferred and an integral part of the experimental design.

As the referee has correctly pointed out, we can directly measure the resistance of the strain-induced barrier and compare it to previously-studied Klein barriers. When biased at pp or nn region (in absence of pn junctions). The 4-probe resistance of the device can be written as R_{tot}

= $R_G + 2R_{cw}$, where R_G is the graphene resistance, R_{cw} is the resistance from a single cavity wall. At high doping (when $n_1 \sim n_2 \sim 10^{12} \text{ cm}^{-2}$), R_{tot} of the cavity device is measured to be $\sim 0.32 \text{ k}\Omega$. At the same doping, R_G is estimated to be $\sim 0.01 \text{ k}\Omega$, from measuring a different region of the same graphene stack without any strain-induced or electrostatically defined barrier. This gives an estimation of $R_{cw} \sim 0.155 \text{ k}\Omega$, similar to that of a Klein barrier defined by pn-junction with known ineffective collimation (or finite Klein angle-dependence) preferred by our device architecture. We have added a new SI section S14 to elaborate these technical points.

2. In the sentence “The absence of these transport features in the nn or pp gate configurations rule out the possibility for them to arise from disorder or Fabry-Pérot interference.”, authors claim that resistance peak appearing only in bipolar gate configurations cannot be due to F-P interference. However, the formation of p-n junction can play a crucial role of giving F-P interference as they can act as a reflector. Indeed, graphene p-n-p or n-p-n junctions routinely show F-P interference. I wonder how F-P interference can be ruled out for giving resistance peaks.

We thank the referee for raising this important point. We have three experimental observations that led us to believe the observation cannot be attributed to F-P.

First, Veselago interference requires a closed loop interference path in the cavity, which exists only when a pn-junction is defined in the cavity and agrees with our experimental observation. The parallel paths of F-P interference, in contrast, should in principle exist for all combinations of carrier types (pp, nn, pn, np) as the referee correctly pointed out, as the two strain-induced barriers exist independently of electrostatic configuration. The observation of the resistance peaks only in the pn or np regime implies that the observation is a consequence of Veselago interference instead of F-P interference.

Second, the location at which F-P interference peaks are found as a function of carrier densities should be in the shape of interference fringes that depend on electron wave vector k , instead of when resonance conditions is met ($|n_1| = |n_2|$, $4|n_1| = |n_2|$ or $|n_1| = 4|n_2|$). The position of observed resistance peaks therefore can't be explained by the F-P mechanism even at a qualitative level.

Third, the novel device architecture is designed for Veselago interference, instead of F-P. The central cavity region containing the interference paths is in between two gates, instead of on top of another central gate. The carrier density in the cavity smoothly changes (as a part of experimental design), instead of homogeneously defined. We therefore do not expect the F-P interference paths to be effectively defined, in a way similar to that of previous work on PNP or NPN junctions.

We have added a new SI section S13 titled “Qualitative Difference Between Veselago interference and Fabry-Perot Interference” and elaborated these important points. We have added “(See SI S13)” to the relevant discussion when we rule out the F-P interference in the main narrative of the paper.

3. In the sentence “Due to the symmetry of the interference loop at $|n_1|=|n_2|$, no bias dependence is observed for the first-order peak (fig. 3e).”, how about the bias dependence for 2nd order peaks ($4|n_1|=|n_2|$ or $|n_1|=4|n_2|$)?

The bias dependence for 2nd and 3rd order peaks is measured and summarized in the SI section S6. Fig. S5b and Fig. S5h show a slightly nonsymmetrical bias dependence for second-order peaks at $|n_1| = 4|n_2|$ in Device 2 and third-order peaks at $|n_1| = 9|n_2|$ in Device 5, respectively. This is

consistent with asymmetric nature of PN junction boundary and the interference paths, when 2nd order resonant condition is met.

4. I failed to follow the logic of the discussion on Fig. 4.

- In the sentence “When $|n_1| < |n_2|$, the uncollimated carriers that escape the cavity predominantly reach voltage probe B, resulting in a transverse voltage proportional to the current density from uncollimated charge carriers (fig. 4b).”, authors discuss the situation for $|n_1| < |n_2|$. However, left circular shape of panel in fig. 4b corresponds to the case for $4|n_1|=|n_2|$ which is the same with Fig. 2f.

We thank referee for raising this important question. The quoted sentence here aims to point out that the refracted angle of charge carriers passing through microcavity is larger than their incident angle, leading them to predominantly reach voltage probe B and thus resulting in a transverse voltage. Fig. 4b is showing a special case of $|n_1| < |n_2|$ at $4|n_1| = |n_2|$, when the second-order Veselago interference further localizes the uncollimated electrons before they escape the cavity. This in turn resulted in reduced transverse voltage at $4|n_1| = |n_2|$, which comes to our response to referee’s insightful comment following this one.

- Authors say transverse voltage would show some voltage for $|n_1| < |n_2|$, but almost zero for $4|n_1| = |n_2|$. However, the condition “ $|n_1| < |n_2|$ ” includes the condition “ $4|n_1| = |n_2|$ ”. How can these two conditions give different results?

We thank referee for raising this valuable question. The experiment is to demonstrate that $4|n_1| = |n_2|$ (with Veselago Interference) is different than any other general $|n_1| < |n_2|$ case (without Veselago interference), in that the carriers are further collimated by the localization effect of Veselago interference. For the general case of $|n_1| < |n_2|$, the uncollimated carriers that escape the cavity predominantly reach probe B and contribute to the measured transverse voltage, except for at $4|n_1| = |n_2|$, when the uncollimated carriers fails to escape the cavity due to the localization effect of 2nd order Veselago interference. The sharp drop in the transverse voltage at (and only at) the $4|n_1| = |n_2|$ (Fig. 4d) demonstrates the added collimation effect from the 2nd order Veselago interference, compared to general $|n_1| < |n_2|$ cases when Veselago interference and its collimation effect are absent. To help emphasize and clarify this important point per referee’s suggestion, we have added the elaboration to the manuscript “**The sharp drop in the transverse voltage at (and only at) the $4|n_1| = |n_2|$ (Fig. 4d) demonstrates the added collimation effect from the 2nd order Veselago interference, compared to general $|n_1| < |n_2|$ cases when Veselago interference and its collimation effect are absent**”.

- I cannot understand the sentence “For the other two interference peaks at $|n_1| = 4|n_2|$ and $|n_1| = |n_2|$, the collimation efficiency is not characterized by the transverse voltage (fig. 4c), as the ballistic carriers (collimated or not) will reach and diffusively scatter from the physical edges of the device before reaching either voltage probe (see SI for more details).” Which part of IS do authors refer to?

We thank referee for going into the details of both the main manuscript and SI. We apologize for not actually including the discussion on this part in SI and cause the confusion. We have added a section in SI S11.

- Why is the outgoing direction of electrons is slightly tilted downward in the left panel of Fig. 4b, while outgoing direction of electrons is perpendicular to the p-n junction barrier in the left panel of Fig. 4c? Shouldn't $|n_1| = 4|n_2|$ and $4|n_1| = |n_2|$ be symmetric?

In our graphene pn-junction where carrier density (or electrostatics) varies along longitudinal direction, the transverse crystal momentum of charge carriers is conserved while the longitudinal counterpart is not. For an asymmetric PN junction such as $|n_1| = 4|n_2|$ and $4|n_1| = |n_2|$, near the two strain-defined cavity walls, the transverse k is the same while the longitudinal k is larger (smaller) at the boundary with higher (smaller) doping, resulting in asymmetric interference loop.

In the cartoon of figure 4a, we showed one particular bias direction where carriers are injected through the cavity wall closest to the source, where electrons with finite incident angles are allowed into the cavity with small Klein-tunneling probability. The injected electrons and their angle-distribution are the same for $|n_1| = 4|n_2|$ and $4|n_1| = |n_2|$ as it is via the strain-defined Klein barrier, independent of electrostatics. However, when the same electron reaches the opposite cavity wall, the incident angle is larger (smaller) for $4|n_1| = |n_2|$ ($|n_1| = 4|n_2|$) due to lower (higher) doping compared to where it is injected. This is what “breaks” the symmetry of the interference loops illustrated in our cartoons where the bias direction is kept the same for the ease of discussion. During the major part of the experiment, an AC bias is applied on top of a zero DC offset, and $|n_1| = 4|n_2|$ and $4|n_1| = |n_2|$ is equivalent. The only exception is figure 4e where a DC bias is intentionally applied to observe the asymmetry expected from the narrative above. We have further elaborated this point in the SI section S9 “Width and DC Bias Dependence of 2nd Order Veselago Interference Peaks”, following referee’s suggestions.

Minor comments:1. Many sentences of the figure captions are repeated in the main text. Please remove redundancy.

We thank referee for asking us to reduce redundancy in the figure captions. We have revised the captions, making it shorter and more precise.

2. Authors should specify which part of SI is referred at every point they are referred.

We thank referee for the important suggestion on improving the readability of the manuscript. We have made changes in the manuscript to specify the corresponding referred part of SI.

3. How was the mobility $\sim 300,000$ cm²/Vs estimated?

We thank the referee for raising this important question. The mobility $\sim 300,000$ cm²/Vs is a rough estimation from observing the half-width of the Dirac peak, compared to that of a typical Hall-bar device fabricated in this lab with similar mobility. We believed that the accurate determination of device mobility is not critical for the physics phenomena described in this work, for as long as the mean-free path is sufficiently longer than the width of the cavity to ensure the ballistic nature of electrons within the interference loops. Unlike a homogeneous Hall bar device, our device architecture is inhomogeneous by design with the presence of strain-defined cavity boundaries. A more quantitative estimation of the carrier mobility can therefore be less straight-forward when using similar methods for Hall bar devices. However, we nevertheless performed new experimental characterization following the referee’s suggestions and provided some elaboration on how to interpret the results in context of the more complicated device architecture. More details are included in the new SI information section S4 “Lower-bound Estimation of Carrier Mobility”.

Reference

1. Kim, S. *et al.* Realization of a high mobility dual-gated graphene field-effect transistor with Al₂O₃ dielectric. *Appl. Phys. Lett.* **4**.
2. Bolotin, K. I. Electronic transport in graphene: towards high mobility. in *Graphene* 199–227 (Elsevier, 2014). doi:10.1533/9780857099334.3.199.
3. Wang, K. *et al.* Tunneling Spectroscopy of Quantum Hall States in Bilayer Graphene p – n Junctions. *Phys. Rev. Lett.* **122**, 146801 (2019).
4. Neil W. Ashcroft , N. David Mermin. *Solid State Physics*, CENGAGE Learning (1976).
5. Ribeiro, R. M., Pereira, V. M., Peres, N. M. R., Briddon, P. R. & Castro Neto, A. H. Strained graphene: tight-binding and density functional calculations. *New J. Phys.* **11**, 115002 (2009).
6. Zabel, J. *et al.* Raman Spectroscopy of Graphene and Bilayer under Biaxial Strain: Bubbles and Balloons. *Nano Lett.* **12**, 617–621 (2012).
7. Kim, S. & Ryu, S. Thickness-dependent native strain in graphene membranes visualized by Raman spectroscopy. *Carbon* **100**, 283–290 (2016).
8. Wang, K. *et al.* Graphene transistor based on tunable Dirac fermion optics. *Proc Natl Acad Sci USA* **116**, 6575–6579 (2019).
9. Elahi, M. M. *et al.* Impact of geometry and non-idealities on electron “optics” based graphene p-n junction devices. *Appl. Phys. Lett.* **114**, 013507 (2019).
10. Sajjad, R. N. & Ghosh, A. W. High efficiency switching using graphene based electron “optics”. *Appl. Phys. Lett.* **99**, 123101 (2011).

REVIEWER COMMENTS

Reviewer #1 (Remarks to the Author):

The authors introduce interference between different incident angles to explain the absence of oscillatory dependence on k . I still don't think this mechanism is sufficient to explain the result. Here are questions regarding my concern.

1) For the plot in Figure R1 or Figure S11, why do the authors sum over k vector? In the real measurement, for a given data point, k is fixed as it depends on the carrier density. What changes is the distance L . However, since the authors always assume small incident angles, L is also relatively constant. Therefore, kL phase is well defined for each data point without any k -averaging. As you vary density along $|n_{-1}| = |n_{-2}|$, there should be oscillatory dependence on k .

2) In S15 where the authors calculated critical magnetic field to break interferences, the authors use average carrier density of 10^{10} cm^{-2} . This selected value is almost 2 order of magnitude lower than the actual density of $7 \times 10^{11} \text{ cm}^{-2}$ used in the measurement of Figure 3d. If the actual density were used, the cyclotron radius would be about 200 nm which is more than enough to form an interference loop. Even if we have to average the density, I would say 10^{11} cm^{-2} which gives cyclotron radius of 70 nm, more than enough to form a loop, is much more reasonable than 10^{10} cm^{-2} . In fact, it would be best if the authors could simulate electron trajectory for linearly varying density inside micro-cavity.

3) What is the reason for a condition to form a second order interference loop to be $|k_{y1}| = 2|k_{y2}|$? It seems to me that this condition relies on the assumption that the charge neutrality line is in the middle of the cavity. However, with $|n_{-1}| = 4|n_{-2}|$, the charge neutrality line will be shifted away from the middle and hence results in a non-closed loop. If the closed loop condition is an important condition to observe the "interference" peaks, I think it's necessary for the authors to quantitatively calculate electron trajectory with varying density to verify the $|k_{y1}| = 2|k_{y2}|$ assumption. In addition, it will provide evidence that there is no closed loop for $|k_{y1}| = a|k_{y2}|$ where a is some other arbitrary numbers.

Reviewer #2 (Remarks to the Author):

I am satisfied with the author's responses and therefore I recommend the manuscript for publication without further changes.

Reviewer #3 (Remarks to the Author):

I thank the authors for responding and answering all the questions faithfully. Now I recommend the publication in Nature Communications.

Responses to Referees' Comments

Reviewer Comments: blue; Responses: Black; Changes Made in Revision: Red; Deletion: Gray
Referee 1

The authors introduce interference between different incident angles to explain the absence of oscillatory dependence on k . I still don't think this mechanism is sufficient to explain the result. Here are questions regarding my concern.

We thank referee's time in reading our rebuttal and providing constructive suggestions. We have carefully addressed each of the comments from the referee, listed point by point below.

1) For the plot in Figure R1 or Figure S11, why do the authors sum over k vector? In the real measurement, for a given data point, k is fixed as it depends on the carrier density. What changes is the distance L . However, since the authors always assume small incident angles, L is also relatively constant. Therefore, kL phase is well defined for each data point without any k -averaging. As you vary density along $|n_{-1}| = |n_{-2}|$, there should be oscillatory dependence on k .

We thank the referee for raising these important points. We have further clarified them in the manuscript and the SI. Specifically:

1. "For the plot in Figure R1 or Figure S12, why do the authors sum over k vector?" We want to clarify that **there is NO summation/integration over k** . We agree with the referee that "for a given data point, k is fixed as it depends on the carrier density", and therefore **we use a single fixed-value k** in our analysis for each data point. Fig. S12b shows k -dependence of several interference loops within the small-incident angle distribution. Fig. S12c **is a summation (of k -dependence) of contribution from these different interference paths, instead of summation over k** . For any given k value in the plot, the summation takes into account the contribution of all interference loop that leads to the measured resistance at data point taken at the said k value. The R versus k plot displays an exponential decay profile, instead of oscillatory dependence on k , as a consequence of the summation over difference interference loops with in the small-incident angle distribution. We have further elaborated these points in the SI section S13 "The measured resistance R at the Veselago interference peaks, is a summation of contribution from all co-existing interference loops (instead of summation over k), which (fig. S12c) shows exponential-decaying profile of k (instead of oscillatory).".

2. "However, since the authors always assume small incident angles, L is also relatively constant. Therefore, kL phase is well defined for each data point without any k -averaging" We would first like to clarify on "small incident angles". The angle distribution of Klein tunneling has been extensively studied in many prior experimental and theoretical works. As the incident angle deviates from perpendicular (zero incidence angle), the probability to enter the cavity (and contribute to Veselago interference) exponentially decays. Therefore, in our data analysis, **we consider carriers with incident angle in the range of $-\theta_0 \sim +\theta_0$** . We want to clarify that **we are NOT referring to a fixed value incident angle ($\theta = \theta_0$)**. **The distribution (despite its small span) has significant consequences** to the qualitative nature of our experimental observation. Specifically,

First, on “**L is also relatively constant**”. k_y continues to change along the loop, while k_x (proportional to θ) does not. The role of k_x in determining the size of interference loops becomes increasing more important towards the center region of the cavity. The **length L of the interference loops can be significantly different** for carriers with incident angle ranging between $-\theta_0 \sim +\theta_0$, despite the small value for θ_0 . To demonstrate this clearly, we have followed the referee’s suggestion on quantitatively simulating the ballistic trajectory of carriers at different incident angles. Figure R1 plots the simulated trajectories of carriers with incident angles being 1° (red), 2° (orange), 3° (yellow), 4° (green), 5° (blue), respectively. We have added this simulation to SI section S13.

Figure R1. Carrier trajectories simulation at different incident angles. Simulations demonstrate the carrier trajectories in the microcavity when the incident angles are 1° (red), 2° (orange), 3° (yellow), 4° (green), 5° (blue), respectively. x and y are the directions that across the junction and along the junction boundary, respectively.

Second, on referee’s note of “**kL phase is well defined for each data point without any k-averaging**”. (1) We want to emphasize that **the normal-distribution nature** of the incident angle (and resulting trajectory) **leads to suppression of oscillatory behavior** (or decoherence of kL phase, no longer well-defined) of each data point, **even with a small standard variation in θ** . We want to clarify that this is consequence of normal distribution (within small incident angle) instead of “**any k-averaging**”. This leads to a decaying profile (instead of oscillatory behavior) shown in fig. S12c. (2) On “**kL phase is well-defined**”, the kL phase, or at least its consequence in transport, is not well-defined in our device architecture, as cross-checked by multiple characterizations within the experiment. Realistic inhomogeneity along strain-defined cavity boundary (location, effective width and height) exists, as characterized in multiple different devices (SI section S6). The inhomogeneity leads to decoherence of the kL term while keep the Veselago interference intact. For example, Veselago interference condition stays independent of cavity width d , while spatial phase sensitively depends on it. **The absence of Fabry-Perot interference** (as a direct prediction of well-defined kL phase) **has also been experimentally verified at the PP and NN regime of multiple devices** (figure 1d and SI section S14). We have further elaborated these important point in the SI section 13.

To sum up the above points, “**Each data point on the interference peak, is contributed by a collection of Veselago interference loops with different individual incident angles and spatial phases, and lack overall coherence (SI S13). Absence of Fabry-Pérot interference (or lack of**

oscillatory dependence on carrier density) is expected and experimentally cross-checked (SI S14).”

We have added this elaboration in the manuscript in addition to the prior-mention expansion of corresponding SI section, to clarify these important points raised by the referee. Future experimental improvement on precisely controlling strain-induced barrier height (that further suppresses angle-distribution) and spatial-homogeneity (that allows more well-defined kL phase for electrons injected at different locations of the cavity) may allow investigation of Fabry-Perot interference in the cavity on top of the Veselago interference. Such experiment goes beyond the scope and central thesis of this work.

2) In S15 where the authors calculated critical magnetic field to break interferences, the authors use average carrier density of 10^{10} cm^{-2} . This selected value is almost 2 order of magnitude lower than the actual density of $7 \times 10^{11} \text{ cm}^{-2}$ used in the measurement of Figure 3d. If the actual density were used, the cyclotron radius would be about 200 nm which is more than enough to form an interference loop. Even if we have to average the density, I would say 10^{11} cm^{-2} which gives cyclotron radius of 70 nm, more than enough to form a loop, is much more reasonable than 10^{10} cm^{-2} . In fact, it would be best if the authors could simulate electron trajectory for linearly varying density inside micro-cavity.

We thank the referee again for carefully reading our revision of SI and raising this valid question. We have added more extensive elaboration on our interpretation of the B -field dependence, with improvement and clarification including:

1. We added clarification that the magnetic field dependence data presented in the paper demonstrates the disappearance of the observed resistance peaks, which **qualitatively agrees** with our physics picture of Veselago **interference thus providing another “sanity-check”**. Quantitative analysis of critical magnetic field of when interference disappears is not possible, nor is it relevant to the central thesis of the work, for the following reasons:

(1) Multiple interference loops with a distribution simultaneously contribute to the observed the resistance peak, **its broadening and eventual disappearance (as a function of B) is**

Figure R2. Magnetic field dependence reproduced in multiple devices. Measured resistance as a function of magnetic field at the carrier density near the interference peak in devices 1, 2, 4 and 5 are presented from panel (a) ~ (d).

smooth, with critical field ill-defined. The Lorentz force changes carrier trajectory and therefore also the Klein refraction probability at the center of the cavity. We do not expect this change to be a sharp transition because: (a) For a given interference loop with known incident angle, the Klein tunneling probability is a continuous function of B . (b) When magnetic field gradually increases, the carriers in the outermost interference loop (with

larger incident angle) will be the first ones that fail to reach the other side of the cavity or

complete the interference loop (than those with smaller incident angle). The field-dependence for each interference loop differs. The measured B -dependence of resistance peak, contributed by a distribution of these loops, is further smoothed.

(2) The magnetic field dependence is qualitatively reproduced in multiple devices, but the quantitative details vary from device to device. We have measured magnetic field dependence of multiple difference devices, as shown in figure R2. The qualitative features, such as broadening of the resistance peak is reproduced in all 4 devices, strengthening our physics picture. The quantitative details of exactly when the peak start to broaden (or where the start of the broadening can even be defined) differs from device to device, as it depends on realistic experimental variations such as smoothness of resulting PN junction, the specific size and uniformity of atomic strain at the boundary of the cavity, even in the state-of-the-art graphene devices.

- 2. We included the simulation of the carrier trajectory, taking into account spatially varying carrier density.** As the referee correctly pointed it out, the spatially varying density inside micro-cavity results in trajectory under B field being much more complicated than typical cyclotron. As elaborated above, the B -field dependence is meant to provide another qualitative supporting evidence (in addition to many others demonstrated in different parts of the paper) that demonstrate qualitative agreement between our interpretation and the observation. Therefore, in the previous version of the paper, to acknowledge the varying carrier density and come up with a straight-forward intuitive argument, we used the carrier density near the center of the cavity ($\sim 10^{10} \text{ cm}^{-2}$, instead of at the boundary where $n \sim 10^{12} \text{ cm}^{-2}$), whose trajectories are by far affected more by the magnetic field and thus determines the overall shape/size of the resulting trajectory. To enhance the qualitative illustration of electron trajectory under B field, instead of this crude argument, we took the referee's suggestion and updated the simulated carrier trajectory at different magnetic field in fig. R3. The simulation takes into account the spatial-varying carrier density, and qualitatively agrees with the observed magnetic field dependence. The detailed discussion of the simulation is included as a new supplementary section in SI S16.

Figure R3. Carrier trajectories simulations under different magnetic field. Simulated carrier trajectories for the first-order Veselago interference path ($|n_1| = |n_2|$), under external magnetic field of 0 T (black), 0.3 T (blue), and 0.5 T (red). The two cavity boundaries are at $y = \pm 25$ nm.

- 3. We have added elaboration of above points in the manuscript,** to clarify the important points noted above. Specifically, we added new narratives “**The magnetic field dependence shows gradual broadening and suppression of observed Veselago interference peak (fig. 3b-d),**

qualitatively agreeing with the expectation and simulated carrier trajectory (fig. 3a, SI 16), and are reproduced in multiple devices (SI S5) with realistic device variations in quantitative details.” to paragraph 9.

3) What is the reason for a condition to form a second order interference loop to be $|k_{y1}| = 2|k_{y2}|$? It seems to me that this condition relies on the assumption that the charge neutrality line is in the middle of the cavity. However, with $|n_{-1}| = 4|n_{-2}|$, the charge neutrality line will be shifted away from the middle and hence results in a non-closed loop. If the closed loop condition is an important condition to observe the “interference” peaks, I think it’s necessary for the authors to quantitatively calculate electron trajectory with varying density to verify the $|k_{y1}| = 2|k_{y2}|$ assumption. In addition, it will provide evidence that there is no closed loop for $|k_{y1}| = a|k_{y2}|$ where a is some other arbitrary numbers.

We thank the referee for raising up these valid questions. We have added more discussions about the condition for Veselago interference to further elaborate these important details.

1. “It seems to me that this condition relies on the assumption that the charge neutrality line is in the middle of the cavity. However, with $|n_{-1}| = 4|n_{-2}|$, the charge neutrality line will be shifted away from the middle and hence results in a non-closed loop.”

We want to clarify that “charge neutrality line being in the middle of cavity” is **NOT a required condition for a close loop interference path**. For 1st order interference peak observed at $|n_1| = |n_2|$, the charge neutrality line is in the center of cavity, merely as a natural consequence of electron-hole symmetric loop. **For the 2nd order interference peak at $|n_1| = 4|n_2|$, the charge neutrality line is 1/5 way in the cavity** (“shifted away from the middle” as referee correctly pointed out). But it does NOT “results in a non-closed loop”, and in fact **is the very condition in forming 2nd order close-loop**. To emphasize this important point, we have made the following elaboration in the revised manuscript: (1) Added more elaboration in relevant supplement section where we discussed about the resonant condition (SI S7-S9). (2) Revised the relevant discussion in main manuscript to emphasize the location of charge neutrality line for 2nd order peaks:

“Similarly, a closed interference loop (bold) can be formed at $|n_1| = 4|n_2|$ and $4|n_1| = |n_2|$, via a sequence of two refractions (at charge neutrality line close to the cavity boundary with lower doping) and three reflections, leading to the second-order Veselago interference peaks” (3) Added simulation of 2nd order interference loop, taking into account of spatially-varying carrier density.

2. “If the closed loop condition is an important condition to observe the ‘interference’ peaks, I think it’s necessary for the authors to quantitatively calculate electron trajectory with varying

Fig. R4 Simulated Carrier Trajectory of Second-order Veselago Interference Loop. Simulated carrier trajectory of 2nd-order Veselago interference loop ($|n_1| = 4|n_2|$). The two cavity boundaries are at $y = \pm 25$ nm. A close-loop interference path forms with the charge neutrality line at $y = 15$ nm, located 1/5 way in the cavity, closer to the boundary with lower doping.

density to verify the $|k_{y1}| = 2|k_{y2}|$ assumption.” Following the referee suggested, we have performed simulation of 2nd order interference loop, taking into account of spatially-varying carrier density, and quantitatively verify that the condition for 2nd order interference peak is indeed $|k_{y1}| = 2|k_{y2}|$. The following revisions have been made: (1) Added the simulation to main manuscript figure 2 to more accurately demonstrate the charge trajectory. (2) Added a new SI section (SI S9), detailing the simulation results and method used.

3. We want to emphasize that **while theoretically, the 2nd order interference peak is expected at exactly $|n_1| = 4|n_2|$, realistic experimental variation exists** from device to device. The exact width and location of interference peaks can depend on realistic device specifics such as homogeneity of electrostatically defined PN junctions and strained-induced barrier, which determines the half-width of interference peak and error bar in determining the peak position (see SI S8, S10). We have reproduced the 1st and 2nd order interference peaks in multiple devices (fig. 2), all in good agreement with expected peak position, within a standard error bar of $\sim \pm 7.4\%$ due to peak broadening from realistic device specifics such as device homogeneity. To emphasize this point, we have added in the manuscript **“The Veselago interferences are reproduced in multiple devices (fig. 2 a-c), with slight variations of microscopic details from realistic device specifics and inhomogeneity (SI S8, S10).”**

Referee 2

I am satisfied with the author's responses and therefore I recommend the manuscript for publication without further changes.

We thank the referee for the time and supporting the publishing of our work.

Referee 3

I thank the authors for responding and answering all the questions faithfully. Now I recommend the publication in Nature Communications.

We thank the referee for the time and supporting the publishing of our work.

REVIEWERS' COMMENTS

Reviewer #1 (Remarks to the Author):

The authors resubmitted the manuscript with additional quantitative analysis on carrier trajectories to address

my comments. Having read the author's responses, I am convinced that the paper is now valid and presents a novel mechanism which produces the observed resistance peaks. I recommend the manuscript for publication.